# The Impact of Ethologically Relevant Stressors on Adult Mammalian Neurogenesis

**DOI:** 10.3390/brainsci9070158

**Published:** 2019-07-04

**Authors:** Claudia Jorgensen, James Taylor, Tyler Barton

**Affiliations:** Behavioral Science Department, Utah Valley University, Orem, UT 84057, USA

**Keywords:** adult neurogenesis, social stress, social isolation, social defeat, dominance hierarchy

## Abstract

Adult neurogenesis—the formation and functional integration of adult-generated neurons—remains a hot neuroscience topic. Decades of research have identified numerous endogenous (such as neurotransmitters and hormones) and exogenous (such as environmental enrichment and exercise) factors that regulate the various neurogenic stages. Stress, an exogenous factor, has received a lot of attention. Despite the large number of reviews discussing the impact of stress on adult neurogenesis, no systematic review on ethologically relevant stressors exists to date. The current review details the effects of conspecifically-induced psychosocial stress (specifically looking at the lack or disruption of social interactions and confrontation) as well as non-conspecifically-induced stress on mammalian adult neurogenesis. The underlying mechanisms, as well as the possible functional role of the altered neurogenesis level, are also discussed. The reviewed data suggest that ethologically relevant stressors reduce adult neurogenesis.


**Highlights:**
Various factors have been identified to influence adult neurogenesis.Chronic social stress due to social isolation or social defeat negatively impacts mammalian adult neurogenesis.


## 1. Introduction

Stress, referring to an aversive or challenging situation experienced by an individual [1], is a ubiquitous part of life, affecting an individual’s physical and psychological well-being. The stimulation of the both the autonomic nervous system as well as the hypothalamic–pituitary–adrenal (HPA) axis characterizes the stress response [2]. One type of aversive or challenging characteristic of the environment in humans originates from persistent or reoccurring social disruption, confrontation, or neglect. Such chronic social stress has been associated with numerous deleterious effects. These effects include higher mortality, increased prevalence of depression, stress-induced cardiovascular responses, delayed wound healing, and amplified inflammatory response, as well as increased cancer mortality [3,4,5,6,7,8,9,10,11]. Interestingly, the absence or lower level of social integration (e.g., unmarried and socially isolated persons) has also been associated with lower psychological and physical health [12]. For example, the prevalence of suicide and the age-adjusted mortality rates in the unmarried are higher than in the married [13]. In addition, unmarried and socially-isolated individuals display higher rates of tuberculosis, deficits in cardiovascular health, and psychiatric disorders like schizophrenia and depression [14,15].

In recent decades, an important line of neuroscience research has focused on the effects of stress on neuroplasticity. It should be noted that research on neuroplasticity is commonly conducted in non-human mammals, since definitive experiments in humans can only be conducted under rare circumstances [16]. Therefore, the ultimate goal is to apply the findings on neuroplasticity obtained from non-human mammals to our understanding of human brain function. In non-human mammals, researchers found that stressful experiences negatively impact structural as well as functional synaptic plasticity. For example, sustained levels of stress cause the retraction and simplification of dendrites in the hippocampus and prefrontal cortex, which, in turn, causes a reduction in synapses [2,17,18]. Stress-induced structural changes also include the reduction of glial cell numbers ([19,20]; but see Reference [21]) and the alteration of dendritic spine density [22]. Long-term potentiation, a measure for functional synaptic plasticity, is also impaired in response to chronic stress (reviewed by Reference [1]). The field—assessing the impact of stress on neuroplasticity—has also developed an appreciation of the fact that stress impacts adult-generated neurons. 

Adult neurogenesis refers to the creation of functional neurons during adulthood by integrating themselves into the existing neuronal circuitry [23]. Adult-generated neurons have been observed in almost all mammals examined so far, including marsupials, moles, rodents, rabbits, sheep, tree shrews, bats, and non-human primates (reviewed in References [23,24]; but see Reference [25]). In addition, evidence exists of adult neurogenesis in humans [16,26,27,28,29]. Most commonly, mammalian adult-generated neurons have been observed in the subventricular zone (SVZ) and the subgranular zone of the dentate gyrus (DG) of the hippocampus, leading to interneurons in the main olfactory bulb and granule cells in the granular cell layer of the DG, respectively [23,30,31,32]. In addition to these traditional brain regions, the presence of adult neurogenesis has also been reported in other non-traditional neurogenic brain structures, including the amygdala (AMY), hypothalamus (HYP), medial preoptic area, neocortex, piriform cortex, and the striatum [24,33,34]. Adult neurogenesis has been recognized as a multi-step process characterized by proliferation (birth of new cells), differentiation (commitment of the newly generated cells to neuronal phenotype often assessed by doublecortin-labeling or BrdU/NeuN double-labeling), migration (traveling of immature neurons to specific location), and maturation/survival (immature neurons display the growth of the dendritic field as well as axon formation, which, in turn, allows them to survive) and functional integration (adult-generated neurons form afferent and efferent connections and display activity-dependent stimulation) [31,35,36,37,38]. Newly generated neurons not recruited into the existing neuronal network are eliminated by apoptosis [39]. While the function of adult-generated neurons is still far from clear, evidence from non-human mammalian studies seems to suggest that these new neurons may play a behavioral role specific to the region in which these neurons reside. Specifically, SVZ adult neurogenesis might play a role in olfactory learning [40,41]. Adult-generated DG neurons might be involved in cognition as well as buffering of the stress response [41,42,43,44]. Furthermore, evidence has emerged that adult-generated neurons may also play a role in reproduction [45,46] as well as the pathophysiology of diseases (such as depression, schizphrenia, epilepsy, Alzheimer’s disease) (reviewed by References [23,47]. Interestingly, the observed functional implications of adult-generated neurons might translate to humans. In particular, cancer treatment, which leads to the blockade of adult neurogenesis, has been associated with deficits in cognitive functions—suggesting that new neurons are involved in cognition [48].

Numerous factors (such as environmental enrichment, exercise such as running, neurotrophic growth factors, and hormones) have been identified to influence the various stages of adult neurogenesis [49,50]. One of these factors that has attracted particular interest has been stress. Indeed, a preponderance of data indicates that stress has a negative impact on adult neurogenesis [51,52,53], and various reviews have focused on discussing several aspects of this link [54,55,56,57]. However, no review to date has specifically discussed the impact of ethologically-related stressors on adult neurogenesis. This can be considered a substantial limitation to the field of neurogenesis. Ethological stressors like psychosocial stress are appealing because they are likely to meet face validity (i.e., they seem to resemble one’s concept of stressful situations), but the importance of focusing on such stressors extends to more concrete concerns. First, although stress is often described as a generalized response, different types of stressors (social versus non-social) can produce qualitatively different behavioral and physiological effects [58,59]. It has even been suggested that the most powerful influences on brain structure and function are due to social influences [60]. Second, some forms of psychosocial stress (e.g., social defeat, intruder paradigm) may suffer from the same potential confound of eliciting nociceptive pain in addition to general stress that occurs with foot shock and some forms of restraint stress; pain does not seem to be integral to the effects of psychosocial stressors, such as position in the social hierarchy [61]. Finally, ethologically-relevant stressors are important for understanding the role of adult neurogenesis by modeling conditions in which an animal has evolved to respond. An understanding of the significance of adult neurogenesis necessitates grasping the interaction between the environment sculpting the brain and behavior influencing neural plasticity [62]. Therefore, a systematic review of ethologically-related stressors might help guide future research on the functional implications of adult-generated neurons. The current review will concentrate on both inter- as well as intraspecies related stressors present in adulthood and their impact on the various stages of mammalian adult neurogenesis. Furthermore, the review will discuss the underlying mechanisms, as well as the possible functional implications of alterations in adult neurogenesis.

## 2. Conspecifically-Induced Psychosocial Stress

Positive social interactions among conspecifics (e.g., strong social bonds between sexual partners and close family members) foster well-being, while negative social interactions (including confrontation and divorce) cause psychosocial stress. A common feature of the animal kingdom is the presence of psychosocial stressors, since animals compete for space, shelter, water, food, and access to mates. The impact of psychosocial stressors can be quite deleterious, because they can impair various functions of biological systems and negatively impact the mental as well as physiological health of an individual [63,64,65]. Here, we will discuss the impact that conspecific psychosocial stressors (induced by the lack or disruption of social bonds and confrontation) have on adult neurogenesis.

### 2.1. Lack or Disruption of Social Interactions

The lack or disruption of social interactions has harmful consequences, leading to various behavioral (e.g., increased anxiety and aggression, learning deficits, facilitation of drug use) [66,67] as well as neuronal changes (e.g., reduction in dendritic spines) [68]. In the laboratory, the lack or disruption of social interactions is often modeled by housing animals separate from other conspecific adults [69]. Using such separation from peers, evidence has emerged suggesting that adult neurogenesis is altered in response to the lack or disruption of social interactions.

Fowler et al. [70] conducted one of the first studies investigating the impact of social isolation on adult-generated neurons in mammals. Sub-chronic (2 day) social isolation in female prairie voles (*Microtus ochrogaster*) augmented the number of newly generated SVZ cells, without altering cell proliferation in the AMY, cingulate cortex, caudate putamen, DG, and HYP. Interestingly, in the same study, chronic (21 day) social isolation appeared to reduce the number of adult-generated cells in the AMY and HYP. However, these changes did not reach statistical significance. No effect of social isolation was noted in the cingulate cortex, caudate putamen, hypothalamus, or main olfactory bulb. The majority of the adult-generated cells in the DG in the sub-chronic condition and the adult-generated cells in the DG and main olfactory bulb in the chronic condition, displayed a neuronal phenotype. Since this report, other studies have assessed the impact of social isolation on adult neurogenesis. Similarly to the increase reported in prairie voles, subchronic (4 day) social isolation increased DG cell proliferation in male, but not female, California mice (*Peromyscus californicus*) [71]. The researchers reported that roughly 40% of the DG cells displayed a neuronal phenotype. Interestingly, sub-chronic (8 day) social isolation did not alter the number of adult-generated DG cells in female or male Wistar rats [72]. Spritzer et al. [73] reported that chronic (15 day) isolation in male Sprague–Dawley rats decreased cell survival in the hilus, but not the granular cell layer, of the DG, with the majority of these adult-generated cells displaying a neuronal phenotype. Furthermore, the researchers found an increase in the number of adult-generated DG cells with a neuronal phenotype, following social isolation [73]. Currently, it is not known why there is an increase in neuronal differentiation and a simultaneous reduction in cell survival. In Wistar rats, chronic (21 day) social isolation reduced cell survival in the granular cell layer of the DG in females, but not males [72]. Similarly, Ruscio et al. [71] observed that chronic (24 day) social isolation reduced DG cell survival only in female, but not male, California mice. In female prairie voles, chronic (42 day) social isolation significantly altered the various stages of adult neurogenesis [74]. Specifically, such long-term isolation reduced cell proliferation in the DG (granular cell layer, hilus, and molecular cell layer) and medial preoptic area; cell survival in the AMY, DG, and ventromedial hypothalamus; and neuronal differentiation in the AMY and DG. Interestingly, social isolation can also influence how other environmental factors impact adult neurogenesis. Typically, running has been shown to increase hippocampal cell proliferation and survival [75,76,77]. However, such running-induced increase in hippocampal cell proliferation and survival disappears in socially isolated rats [78,79].

The disruption of social interactions can also be modeled by repeated separations of a parent from their offspring [80,81]. This has most commonly been studied by separating females from their offspring. Unfortunately, only one study to date has assessed the impact of maternal separation on adult neurogenesis. Similarly to the effect of the separation from peers, the repeated separation from the offspring (6 h per day for 14 consecutive days, referred to as intermittent in the Appendix A) resulted in the reduction of DG cell proliferation and a simultaneous increase in DG apoptosis in rat mothers [82].

Overall, the presented data seem to indicate that the lack or disruption of social interaction alters mammalian adult neurogenesis in a time- and sex-specific way. Specifically, sub-chronic (2 and 4 day) social isolation seems to augment the number of adult-generated cells, while chronic (15, 21, 24, and 42 day) social isolation seems to reduce cell survival. One data point suggests that 8 days might possibly be the length at which social isolation impact switches from up- to down-regulation [72]. Additionally, the data might suggest that females are more susceptible than males to the impact of social isolation, as two reports [71,72] suggest. While such a sex difference seems plausible (see supporting behavioral evidence in Reference [83]), it should be acknowledged that any generalizations ought to be viewed with caution, due to various limitations. One aspect of concern is the extremely low number of studies. Two of these studies only assessed females, making it difficult to see sex-specific patterns. Further, three studies investigated socially monogamous and bi-parental species (specifically prairie voles and California mice), while the other four studies focused on non-monogamous species (specifically, Wistar and Sprague–Dawley rats), making it unlikely that species-specific patterns could be discerned. Additionally, there are methodological differences between studies (including the methods to assess the stages of adult neurogenesis, as well as the type of control group used). 

As few studies have investigated the impact of social isolation on adult neurogenesis, relatively little is known about the underlying mechanisms. Social isolation rearing (social isolation that starts at the time of weaning) has been shown to cause various endocrinological changes. Such alterations include the release and turnover rates of catecholamines and serotonin, as well as gene expression for oxytocin and vasopressin, and cause modifications in gene expression for oxytocin and vasopressin [84,85,86]. Furthermore, social isolation in adulthood causes the release of oxytocin [87]. Unfortunately, researchers to date have not yet investigated what the underlying mechanism is for the isolation-induced changes in adult neurogenesis. Therefore, it is plausible to speculate that the endocrinological changes mentioned above might play a role in the regulation of isolation-induced changes of adult neurogenesis. Future research should systematically add studies to assess whether isolation-induced alterations of adult neurogenesis are indeed time- and sex-specific. Furthermore, researchers should investigate the potential underlying mechanisms, as that might guide exploration of the functional implications of isolation-induced plasticity. 

### 2.2. Confrontation

Similarly to the deleterious effects of social isolation, confrontation (e.g., fighting, divorce) causes numerous adverse behavioral, physiological, and neuronal changes. Examples of behavioral changes include reduced aggression and defense, as well as increased anxiety following social stress due to confrontation (reviewed by Reference [58]). Social-stress-induced physiological changes include activation of the HPA axis, as well as increased adrenal weight. Neuronal changes in response to social stress consist of alterations in various neurotransmitter systems (including serotonin, dopamine, and vasopressin), reduction in dendritic branching and dendritic length, and impacted adult neurogenesis [58,88]. Confrontation has been modeled by two specific paradigms in the laboratory: dominance hierarchies and social defeat [58]. Here, we will investigate the literature that addresses the impact of dominance hierarchies and social defeat on the various stages of mammalian adult neurogenesis.

#### 2.2.1. Dominance Hierarchies

Competition for resources such as space, food, and water, as well as potential mates, occurs frequently across most of the animal kingdom. The competition-induced activation of the HPA axis seems to occur commonly among animals. HPA activity might be influenced by social status and rank, because individuals might differ in the number of resources they are able to access. Specifically, dominant individuals tend to have preferential access to resources, in comparison to lower status individuals [89].

While dominance hierarchies can be commonly observed in the animal kingdom, only a small number of studies has assessed the impact of dominance on adult neurogenesis. Kozorovitskiy and Gould [61] observed that dominant male Sprague–Dawley rats displayed increased DG cell survival in comparison to control and subordinate males. Authors noted that the majority of DG cells displayed a neuronal phenotype. Interestingly, DG cell proliferation, neuronal differentiation, and commonly used indices of stress (such as corticosterone levels, ratio of adrenal gland to body weight ratio, and ratio of thymus gland to body weight ratio) did not differ across groups. One possible explanation for the lack of dominance-induced impact was the sub-chronic duration of exposure to the dominance hierarchy (3 days). Using a chronic exposure period (6 weeks) altered the rate of DG cell proliferation [90]. Specifically, dominant male Sprague–Dawley rats displayed 35% more cell proliferation than subordinate rats. Unfortunately, none of the other stages of neurogenesis (such as differentiation) were evaluated. Similarly to the findings in rats, dominant male baboons (*Papio cynocephalus anubis)* displayed a greater DG cell proliferation than subordinate baboons [91]. In addition, the dominant male baboons (hierarchy status remained stable during the existence of the colony) displayed a higher number of immature DG neurons compared to the subordinate baboons. Based on these studies, it seems that dominance enhances, while subordinance suppresses, adult neurogenesis. Interestingly, the reverse pattern appeared when investigating a eusocial animal, the Damaraland mole-rat (*Cryptomys damarensis*) [92]. In this species, the subordinate female animals (reproductively suppressed worker mole rats) displayed the highest level of hippocampal cell proliferation and highest number of immature neurons, while the dominant female animals (actively breeding queens) displayed the lowest level of hippocampal cell proliferation and lowest number of immature neurons [92,93].

Overall, it seems that dominance positively impacts the stages of hippocampal adult neurogenesis, while subordinance negatively impacts the stages of hippocampal adult neurogenesis. Future research should systematically investigate whether this pattern is true for other non-eusocial animals and reversed for eusocial animals. In addition, researchers should investigate other brain regions in addition to the hippocampus, and analyze the underlying mechanism of dominance-induced alterations of adult neurogenesis. At this point, due to the small number of studies investigating this topic, very little is known about the underlying mechanism. As dominance and subordination are likely associated with aggression, it is plausible that aggression, as well as regulators of aggression (such as arginine vasopressin, oxytocin, serotonin, and testosterone [94,95,96]), may play a role in the confrontation-induced changes of adult neurogenesis.

#### 2.2.2. Social Defeat

Social defeat is another paradigm to study confrontation between conspecific animals. In the laboratory, this is commonly studied by having a resident defend its home cage against an unfamiliar same-sex intruder, which leads to the defeat of the intruder. This powerful psychosocial stressor dramatically changes physiology (e.g., HPA activation [97]), neuroanatomy (e.g., reduction in dendritic branching and neuron number [98]), and behavior (including deficits in social interactions and an increase in anxiety; reviewed by Reference [99]). In addition, social defeat has been documented to alter adult neurogenesis in a variety of mammalian species. 

One acute social interaction with a dominant same-sex conspecific led to the reduction of DG cell proliferation of defeated male tree shrews (*Tupaia belangeri*) and defeated male common marmoset monkeys (*Callithrix jacchus*), in comparison to unstressed control animals [51,100]. The authors reported that the majority of these cells displayed neuronal morphology. Interestingly, one acute social defeat exposure did not alter hippocampal cell proliferation in male CFW mice or Sprague–Dawley rats [101,102]. The observed variances in the impact of one acute social defeat exposure on DG cell proliferation across studies might in part be due to differences in the social defeat paradigms. In particular, the social defeat duration in the studies using rodents (mice: 5 min direct exposure, rat: 20 min direct exposure) was significantly shorter than the social defeat duration in the studies using tree shrews and marmosets (1 h direct exposure). While one acute social defeat in male Sprague–Dawley rats did not alter DG cell proliferation, it did reduce the survival of newly generated DG cells in subordinate Sprague–Dawley rats [102]. 

Similarly to the impact of one acute social defeat exposure, sub-chronic social defeat (daily defeat for 5 consecutive days) reduced the number of immature DG neurons in male Wistar rats in comparison to non-stressed controls [103,104]. Five consecutive days of defeat stress reduced DG cell survival in Wistar rats without altering differentiation [104]. The majority of DG neurons were observed to express a neuronal phenotype. Following sub-chronic (repeated daily for 7 and 10 consecutive days) social defeat stress, male C57BL, C57BL/6, and CD1 mice also showed a reduction in DG cell proliferation, survival, and differentiation [105,106]. No change in AMY cell proliferation was noted [105]. Chronic social defeat stress also altered adult neurogenesis. Specifically, chronic social defeat lowered DG cell proliferation in male Wistar rats and tree shrews, as well as lowering DG cell survival in male Wistar rats [107,108,109,110,111]. The majority of adult-generated DG cells in male Wistar rats displayed a neuronal phenotype, and chronic social stress did not seem to alter neuronal differentiation [108,110]. Cell proliferation was the only neurogenesis stage that was evaluated in tree shrews [107,109,111]. It should be noted that the psychosocial stress-induced reduction in DG cell proliferation was age-dependent—the oldest tree shrews showed the greatest vulnerability to the defeat stress [111]. Additionally, chronic psychosocial stress lowered cell proliferation and cell survival in the prefrontal cortex in male Wistar rats, without altering cell proliferation or survival in the subventricular zone and primary motor cortex [110]. The majority of cells in the prefrontal cortex displayed a glial phenotype, and chronic social stress did not alter neuronal differentiation of adult-generated cells [110].

Overall, the research seems to indicate that acute, subchronic, and chronic exposure to defeat stress negatively impacts hippocampal adult neurogenesis. Unfortunately, most studies have only assessed the impact of social defeat on cell proliferation, rather than also assessing the impact on other neurogenic stages. Therefore, future studies should systematically investigate the effect of social defeat on the various stages of adult neurogenesis. Furthermore, future research needs to assess whether females are also susceptible to social defeat stress—a topic that has remained vastly understudied. Lastly, researchers have primarily focused on the DG. Very few researchers have looked at the impact of social defeat on other neurogenic brain regions. Future research should systematically investigate whether defeat stress impacts adult-generated neurons outside the hippocampus. 

As few studies have investigated the impact of social defeat on adult neurogenesis, relatively little is known about the underlying mechanisms. An inverse relationship was noted between the number of newly generated hippocampal cells and the total number of received bites [101]. In addition, Mitra et al. [105] observed that the defeat-induced reduction of adult-generated cells was related to the frequency of displayed defensive behaviors. One hypothesis about the underlying mechanism is that the testosterone reduction following social defeat might play a role in lowering adult neurogenesis. Interestingly, testosterone levels are reduced in response to repeated social defeat exposure, but the social defeat-induced alteration of adult neurogenesis is not related to the altered levels of testosterone [104]. Therefore, future research needs to investigate other factors that might play a role in the social stress-induced impact on adult neurogenesis. Other potential neural correlates of defeat-induced reduction of adult-generated neurons include vasopressin and oxytocin, since both neuropeptides showed alterations in response to social defeat [112]. 

## 3. Nonconspecifically-Induced Psychosocial Stress

In addition to conspecifically-induced psychosocial stress, interactions with non-conspecifics can also lead to psychosocial stress. Specifically, prey animals such as rats and mice experience stress in response to exposure to a predator. Indeed, it has been well-documented that even the exposure to the odor of the predator alone leads to the display of innate defensive responses in prey animals, making the use of predator odors a highly suitable model to study stress in response to ethologically relevant stimuli [113,114]. The three most commonly used predator-related odors are cat odor, ferret odor, and trimethylthiazoline (TMT, a major component of fox feces) [113]. It should be noted that a limited number of studies have been conducted to assess the impact of non-conspecific stressors on adult neurogenesis. 

Specifically, 1 h of TMT exposure significantly reduced cell proliferation in the rat DG [115,116,117], while a brief (20 min) exposure did not alter rat DG cell proliferation [118]. Interestingly, this TMT-induced reduction in cell proliferation was only observed in male, but not female, rats [116]. In addition to the impact of TMT on cell proliferation, predator odor also temporarily reduced DG cell survival. One week after TMT exposure, the number of surviving DG neurons was reduced, a difference that was no longer observed 3 weeks later [115]. Neuronal differentiation was not impacted by 1 h TMT exposure [115,116].

Overall, nonconspecifically-induced psychosocial stress seems to negatively alter adult neurogenesis in a sex- and time-specific manner. Specifically, males seem to be more susceptible than females, and the stressor exposure needs to be of sufficient length (longer than 20 minutes). As few studies have investigated the interaction between predator stress and adult neurogenesis, relatively little is known about the underlying mechanism. However, it is plausible to speculate that changes in adrenal steroids (e.g., corticosterone) in response to the exposure to predator-related stimuli might in part regulate the observed changes in adult neurogenesis [119]. In addition to the need to investigate the underlying mechanism of predator-induced alterations, future studies also need to assess the functional implications of these changes.

## 4. Conclusions and Future Direction

Adult neurogenesis is an ongoing function of the brain that has been linked to a variety of functions, including learning and memory [120,121,122,123,124,125] and emotional regulation [41,42,43,44], as well as playing a role in the pathophysiology of diseases [23,47]. Most investigations indicate a negative correlation between adult neurogenesis and stress, particularly intense or long-term stress [51,52,53,54,55,56]. Acute and sub-chronic stress exposure has resulted in mixed effects, with some studies showing an increase and others showing no alteration in hippocampal neurogenesis [51,100,101]. On the contrary, long-term chronic stress has consistently been found to be negatively correlated with adult neurogenesis, a relationship that may be linked with some of the detrimental effects of chronic stress [51,52,53,54,55,56]. However, the complex relationship between stress and neurogenesis is an active area of neuroscience research, and there are many questions still to be resolved.

As mentioned above, caution should be exercised in generalizing many of the findings from the research discussed here. Most studies only assessed the impact of psychosocial stressors on cell proliferation, rather than exploring subsequent neurogenic stages. The importance of up- or down-regulation of cell proliferation could be better understood by tracking the survival and integration of the new cells. Additionally, researchers have primarily focused on adult neurogenesis within the hippocampus. Evidence for adult neurogenesis has also been found in other regions, such as the amygdala, hypothalamus, and neocortex [33,34,126]. However, few studies have examined the impact of psychosocial stressors such as isolation and social defeat on newly generated cells outside the hippocampus (see Appendix A). Future research should systematically investigate changes in non-traditional neurogenic regions, particularly in areas associated with coordinating or mediating stress and learning, such as the amygdala and hypothalamus.

Another critical gap in our current knowledge is an understanding of possible sex-related differences in the relationship between stress and neurogenesis. Male subjects are more commonly represented in this literature, and few studies have included both male and female subjects, despite well-established differences in stress-related behavior. Indeed, structural or neurochemical sexual dimorphism has been reported in relevant regions of the brain, including the hippocampus, hypothalamus, and other nuclei associated with emotional processing (e.g., bed nucleus of the stria terminalis) [127,128,129,130,131]. Systematic investigation that includes male and female subjects may also help to explore the mechanisms mediating the relationship between stress and neurogenesis through looking at the influence of androgens such as testosterone.

The developmental age of animal models is another important factor not to be overlooked. Studies exploring the relationship between stress and neurogenesis in rats frequently refer to subjects as being adult animals, but the age of subjects varies considerably across studies. Subjects are often as young as 40 days [132,133]—an age that is often considered analogous to adolescence in humans rather than true maturity or adulthood [134,135]. Other studies utilized rats well into adulthood at 70 days or more [72,77,136], and still others report subjects as being adult, but do not specify age [52,61,78,90,137]. The age disparity can be particularly problematic when investigating psychosocial stressors, as there are likely to be differences in behavioral responses to these stressors based on developmental age.

Another limitation is that the majority of adult neurogenesis research to date has focused on non-ethological stressors. Some stressors, such as foot shock or extended restraint stress, are useful when exploring basic processes in a laboratory setting, due to the level of experimental control allowed to a researcher. However, it is unlikely that most humans and wild non-human animals will experience some of these stimuli, and so ethological stressors can be valuable in generating findings that are more generalizable outside of the laboratory. Many animals are likely to experience the ethological stressors associated with potentially agonistic social encounters or disruptions in social connections. The presence of these psychosocial stressors, particularly when chronic (i.e., repeated or long-term), has long been associated with poor health outcomes, and converging evidence suggests that such stress reduces aspects of neurogenesis.

Psychosocial stressors can include the disruption of social contact as well as agonistic encounters with conspecifics or predators. For social animals, prolonged disruptions in social interaction can negatively impact neurogenesis, although the nature and extent of this impact may vary with factors such as age, sex, and species [72,73,74,138]. Isolation from age-matched conspecifics and repeated isolation from offspring reduces new cell proliferation or survival in the hippocampus [82], a finding consistent with other studies highlighting volumetric and morphological changes in the ventral hippocampus during chronic stress [139,140,141,142]. Chronic and aversive social encounters alter neurogenesis as well. With the possible exception of eusocial animals, established position within a dominance hierarchy is associated with differences in neurogenesis, with dominant animals generally exhibiting enhanced neurogenesis compared to subordinate members of the social group [90,91]. Truly agonistic encounters with conspecifics, as modeled by repeated social defeat, show a similar pattern of altering the process of hippocampal neurogenesis, in that the defeated animals exhibit lower cell proliferation or cell survival [101,103,106]. The alterations in hippocampal neurogenesis observed with non-conspecific social stressors are observed when a subject is exposed to a predator or to the odors of the predator. As with conspecific social stressors, this decrease is consistently observed only when the subject has been exposed to the stressor for extended periods of time and at a significant intensity [115,116,117]. 

In sum, adult neurogenesis seems to be negatively correlated with intense or long-term psychosocial stress. However, the full implications of this relationship have yet to be realized. In addition to the limitations already discussed, future work is needed to determine the long-term effects of the depressed neurogenesis observed following psychosocial stressors. Although there is likely to be a causal association between chronic psychosocial stress and decreased neurogenesis, the link between alterations in neurogenesis and the poor health outcomes associated with such stressors has not yet been thoroughly explored. Indeed, the functional role of adult-generated neurons remains an active area of investigation. However, due to the likely involvement of hippocampal neurogenesis in long-term memory formation, as well as the positive correlation between neurogenesis, relief from anxiety and depression-like symptoms [108], and an increased positive emotional state [143], it is likely that reduced neurogenesis may be implicated in the pathological response to chronic stress. Future research into the role of neurogenesis in our physical and psychological responses to stress, particularly psychosocial stressors that can be more easily generalized to daily life, may therefore help address the serious health concerns associated with chronic stress.

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
