# Peer review of "The Impact of Ethologically Relevant Stressors on Adult Mammalian Neurogenesis"

_brainsci, 2019, doi:10.3390/brainsci9070158_

Round 1
Reviewer 1 Report
General comment: This is a nice review on the impact of stressors on adult neurogenesis by Jorgensen and colleagues. I have some comments.
Line 42: The author stated that “Stress-induced structural changes also include the reduction of the number of glia”. However, my understanding is that some studies have reported that stress increases the number of glia as well as their activity levels. These points should be mentioned as well.
Line 50-51: It would be nicer if authors could also mention the role of adult neurogenesis in disease.
Line 67-70: I feel that the author should mention relationship between adult neurogenesis and neurodegenerative disease in human to make the paragraph more persuasive for readers.
Line 97-131: I would like to see a summarized table here for the better understanding (e.g. species, stress type and duration, sex, method for confirming neurogenesis).
Line 176-: What is the definition of “dominant” in this review? Please explain.
Line 239-241: The use of words “neuronal” and “cell” should be clearly distinguished throughout the manuscript to avoid misleading the readers. In ref.96, Czéh B et al. reports that “they found no evidence of neurogenesis in the mPFC, instead the newly generated cells differentiated mainly into NG2-positive glia and to a minor portion into endothelial cells”. In this review, the authors use the term “cell proliferation or survival”. Does it indicate “neurogenesis” or not? There are similar points in other sections. Please make these points clear.
Line 247: The author stated “no study to date has investigated females”. This sentence is not clear to me. Harris AZ et al., have reported SDS model in female mice (Harris AZ et al., A Novel Method for Chronic Social Defeat Stress in Female Mice. Neuropsychopharmacology. 2018 May;43(6):1276-1283.)
Line 252-261: Is there any other candidates other than testosterone?
Line 279: It should be noted that Zhao H et al. used an adult cat as a predator stimulus. Thus,
it would not be proper to compare this study, which should include not only an odor stimulus but also several stimuli, with experiments using predator odor (TMT).
Author Response
We appreciate the very positive and constructive comments from the reviewers and have revised the manuscript accordingly. Details of the changes are outlined below.
REVIEWER 1
Comments and Suggestions for Authors
General comment: This is a nice review on the impact of stressors on adult neurogenesis by Jorgensen and colleagues. I have some comments.
1. Line 42: The author stated that “Stress-induced structural changes also include the reduction of the number of glia”. However, my understanding is that some studies have reported that stress increases the number of glia as well as their activity levels. These points should be mentioned as well.
Response: This has been corrected.
2. Line 50-51: It would be nicer if authors could also mention the role of adult neurogenesis in disease.
Response: Thank you greatly for the suggestion. This role of adult neurogenesis has been mentioned (on page 4 first paragraph and on page 9 first paragraph of the conclusion).
3. Line 67-70: I feel that the author should mention relationship between adult neurogenesis and neurodegenerative disease in human to make the paragraph more persuasive for readers.
Response: Thank you greatly for the suggestion. This role of adult neurogenesis has been mentioned (on page 4 first paragraph).
4. Line 97-131: I would like to see a summarized table here for the better understanding (e.g. species, stress type and duration, sex, method for confirming neurogenesis).
Response: Thank you greatly for the suggestion. A table has been added.
5. Line 176: What is the definition of “dominant” in this review? Please explain.
Response: Dominance refers position in a social hierarchy. This conceptual variable is variably operationalized based on the species of animal in question and constraints created by the study or laboratory in question. Generally speaking, a dominant individual is one that establishes greater access to resources (e.g., food or mating opportunities), although this is often indirectly measured by relative success in agonistic encounters between one of more (usually male) individuals. Some of the studies referenced in this review employ different methods for assessing (or establishing) dominance, even within the same species. We accept that there are likely to be some differences in the psychosocial stress generated under these different paradigms, particularly when contrasting paradigms that do or do not include physical aggression.
6. Line 239-241: The use of words “neuronal” and “cell” should be clearly distinguished throughout the manuscript to avoid misleading the readers. In ref.96, Czéh B et al. reports that “they found no evidence of neurogenesis in the mPFC, instead the newly generated cells differentiated mainly into NG2-positive glia and to a minor portion into endothelial cells”. In this review, the authors use the term “cell proliferation or survival”. Does it indicate “neurogenesis” or not? There are similar points in other sections. Please make these points clear.
Response: Thank you for pointing out this lack of clarity. We have added and/or changed wording to make this distinction clearer.
7. Line 247: The author stated “no study to date has investigated females”. This sentence is not clear to me. Harris AZ et al., have reported SDS model in female mice (Harris AZ et al., A Novel Method for Chronic Social Defeat Stress in Female Mice. Neuropsychopharmacology. 2018 May;43(6):1276-1283.)
Response: The wording has been changed to state that very few studies have investigated social defeat in females.
8. Line 252-261: Is there any other candidates other than testosterone?
Response: We have added two other possible candidates, namely vasopressin and oxytocin.
9. Line 279: It should be noted that Zhao H et al. used an adult cat as a predator stimulus. Thus, it would not be proper to compare this study, which should include not only an odor stimulus but also several stimuli, with experiments using predator odor (TMT).
Response: Thank you for pointing this out. To make the comparisons easier, we have deleted this reference.
Reviewer 2 Report
Review of “The impact of ethologically relevant stressors on adult mammalian neurogenesis” by Jorgensen, et al. for Brain Sciences.
The manuscripts reviews literature published specifically about the impact of ethologically relevant stressors on adult neurogenesis. The authors start by describing stress and indicating the effect it has on neural plasticity. The authors briefly describe neurogenesis and associated terminology. The paper is organized by type of stressor. First the effects of social stressors such as social defeat, maternal separation, and social isolation on neurogenesis are described. Then the effects on non-social ethologically relevant stressors on neurogenesis are briefly described. Overall, ethologically relevant stressors reduce neurogenesis, similar to what is found using other non-ethologically relevant stressors. A few specific future directions are discussed, such as looking outside the hippocampus and investigating sex differences. Furthermore, function of these new neurons in the brain remains elusive.
This is a one-of-a-kind specific topic for a review paper. Only one other review is similar to this topic (Social regulation of adult neurogenesis: A comparative approach. Holmes MM Front Neuroendocrinol. 2016 Apr;41:59-70. doi: 10.1016/j.yfrne.2016.02.001), but does not touch on the nonconspecific predator odor types of stress. Another very recent review focuses on social stressors, but does not focus on neurogenesis (Rodent models of social stress and neuronal plasticity: Relevance to depressive-like disorders. Patel, et al. Behav Brain Res. 2019 Sep 2;369:111900. doi: 10.1016/j.bbr.2019.111900). The writing is almost always excellent. It is well organized throughout, but within the sections some improvements on organization could be made (see specific comments). An extensive review of the literature is included in the manuscript. However, some additional references are suggested (see specific comments). Several times review articles are cited where I would expect to see original research cited (e.g., line 156 and 220). I would suggest expanding on what is special about ethologically relevant stressors. Perhaps mention face validity as Patel et al. 2019 do. I would also suggest expanding the discussion to compare and contrast how ethologically relevant stressors are different from those observed following general laboratory stressors with regard to neurogenesis. I suggest the conclusion be added to the abstract. Adding a table summarizing all of the results reviewed would be useful. I would suggest sorting the table by type of stressor to match the organization of the manuscript. As evidenced from the title, this manuscript focuses on adult neurogenesis. I suggest including a brief description of the ontogeny of neurogenesis.
Specific comments:
Line 16, add a general conclusion to the abstract
Line 51, are there any more recent articles suggesting neurogenesis occurs in humans?
Lines 57-61, perhaps expand on this neurogenesis description, define integration as well
Line 63, remove “in” at end of sentence.
Line 75, change references to “[45-48]”
Lines 78-82, perhaps add more information about why ethologically relevant stressors are useful in the laboratory.
Lines 80-82, the sentence starting with “lastly” does not appear to be related to the first two points regarding ethologically relevant stressors.
Lines 100 and 102, While the reference chosen, Cacioppo, et al. 2015 [57], to support the author’s assertion that social deprivation activates the HPA axis does indeed make this claim in its text, I recommend avoiding this reference as support. First of all, it is not an original research article. Also, whether or not social isolation housing causes HPA hyperactivity is a very controversial topic with many contradicting original research findings. Another review article cited later in your manuscript confirms this. From Mumtaz, et al. 2018 [72], “…effects of social isolation on HPA axis…in rats are not consistent.” The problem I have with the author’s chosen reference (i.e., [57]) is that the cited original research in Cacioppo, et al. 2015 does not actually support the statement and even contradicts what is reported in a table from that very same review article (i.e., [57]). Cacioppo, et al. 2015 write, “Studies in rats similarly suggest that chronic social isolation increases corticosterone levels when experimental animals are socially isolated from a group of same-sex rats (Djordjevic et al. 2010; Dronjak et al. 2004; Garrido et al. 2012; Zlatkovic ́ & Filipovic ́ 2012, 2013), but inconsistencies have also been observed (cf. Pournajafi-Nazarloo & Partoo 2011).” This statement is followed by a general conclusion that even appears in the abstract. The problem is that only one reference cited actually found increased corticosterone levels in isolated animals (i.e., Djordjevic, et al. 2010). They also fail to discuss how the corticosterone measurement methods differed between the studies. I recommend supporting your writing with more carefully constructed references.
Lines 105-131, While the duration of isolation is important, especially considering that brief episodes of isolation activate the HPA axis whereas prolonged isolation housing does not necessarily, it is also important to discuss the age during which isolation occurs. Are there any differences between the impact of isolation rearing (adolescence) versus isolation housing (adulthood) on neurogenesis?
Line 131, you could add another reference here: Kozareva, et al. 2010, Hippocampus PMID: 28972669
Line 156, I would expect to see original research cited here.
Line 220, I would expect to see original research cited here instead of [42].
Line 239, Is “long-term psychosocial stress” the same as chronic or subchronic social defeat? These terms are confusing. With so few references it might be easier to just put the details of the social defeat methodology in the manuscript, or perhaps separately in the table that I recommend adding.
Line 247, I challenge the author’s no one has investigated this statement. These references investigate neurogenesis following female social defeat contrary to the author’s assertion: Mol Psychiatry. 2010 Dec;15(12):1152-63. doi: 10.1038/mp.2010.34. Epub 2010 Mar 23.
Environmental enrichment requires adult neurogenesis to facilitate the recovery from psychosocial stress.
Schloesser RJ1, Lehmann M, Martinowich K, Manji HK, Herkenham M.
Horm Behav. 2015 Apr;70:30-7. doi: 10.1016/j.yhbeh.2015.01.010. Epub 2015 Feb 25.
Exposure to social defeat stress in adolescence improves the working memory and anxiety-like behavior of adult female rats with intrauterine growth restriction, independently of hippocampal neurogenesis.
Furuta M1, Ninomiya-Baba M2, Chiba S2, Funabashi T3, Akema T3, Kunugi H2.
Lines 262-289, Section 3 appears to only review the effects of TMT stress on neurogenesis. This immediately following a statement that most of the research out there is about predator-related odors. Where is the review of the more common type of non-conspecific psychosocial stress?
Line 278, which part of the previously reviewed stages of neurogenesis includes this measure of apoptotic neurons.
Line 293, this is the first mention of “integration” since the introduction. Are differentiation maturation, and migration not important for “species-typical behavior”? Perhaps include an example of a species-typical behavior. Consider rewriting.
Lines 295-97, consider deleting text from particulary to resolved as these words do not add anything to the paragraph and detract from the main point.
Lines 322-330, these lines were particularly useful. I would like to see more of this in the manuscript.
Line 342, rewrite “social can”. Social what?
Line 347, non-appetitive? consider a different word such as aversive or neutral. Consider rewriting.
Line 350, agonist is a noun. In my opinion this is a poor word choice when you need an adjective to describe social defeat. Consider rewriting.
Line 354, consider rewriting, “exposed to a predator or related stimuli such as odor” to be more clear.
Lines 290-371, Consider writing about what is different about the conclusions drawn from research on ethologically relevant stressors and ordinary laboratory stressors regarding neurogenesis, and even other topics. For instance, social stressors usually intensify drug-seeking behavior, whereas non-social stressors often do not.
Author Response
We appreciate the very positive and constructive comments from the reviewers and have revised the manuscript accordingly. Details of the changes are outlined below.
REVIEWER 2
Comments and Suggestions for Authors
Review of “The impact of ethologically relevant stressors on adult mammalian neurogenesis” by Jorgensen, et al. for Brain Sciences.
The manuscripts reviews literature published specifically about the impact of ethologically relevant stressors on adult neurogenesis. The authors start by describing stress and indicating the effect it has on neural plasticity. The authors briefly describe neurogenesis and associated terminology. The paper is organized by type of stressor. First the effects of social stressors such as social defeat, maternal separation, and social isolation on neurogenesis are described. Then the effects on non-social ethologically relevant stressors on neurogenesis are briefly described. Overall, ethologically relevant stressors reduce neurogenesis, similar to what is found using other non-ethologically relevant stressors. A few specific future directions are discussed, such as looking outside the hippocampus and investigating sex differences. Furthermore, function of these new neurons in the brain remains elusive.
This is a one-of-a-kind specific topic for a review paper. Only one other review is similar to this topic (Social regulation of adult neurogenesis: A comparative approach. Holmes MM Front Neuroendocrinol. 2016 Apr;41:59-70. doi: 10.1016/j.yfrne.2016.02.001), but does not touch on the nonconspecific predator odor types of stress. Another very recent review focuses on social stressors, but does not focus on neurogenesis (Rodent models of social stress and neuronal plasticity: Relevance to depressive-like disorders. Patel, et al. Behav Brain Res. 2019 Sep 2;369:111900. doi: 10.1016/j.bbr.2019.111900). The writing is almost always excellent. It is well organized throughout, but within the sections some improvements on organization could be made (see specific comments). An extensive review of the literature is included in the manuscript. However, some additional references are suggested (see specific comments). Several times review articles are cited where I would expect to see original research cited (e.g., line 156 and 220). I would suggest expanding on what is special about ethologically relevant stressors. Perhaps mention face validity as Patel et al. 2019 do. I would also suggest expanding the discussion to compare and contrast how ethologically relevant stressors are different from those observed following general laboratory stressors with regard to neurogenesis. I suggest the conclusion be added to the abstract. Adding a table summarizing all of the results reviewed would be useful. I would suggest sorting the table by type of stressor to match the organization of the manuscript. As evidenced from the title, this manuscript focuses on adult neurogenesis. I suggest including a brief description of the ontogeny of neurogenesis.
Specific comments:
1. Line 16, add a general conclusion to the abstract
Response: Minor changes in the abstract have been made and a general conclusion was added to the abstract.
2. Line 51, are there any more recent articles suggesting neurogenesis occurs in humans?
Response: Two additional references have been added (namely Liu et al. 2008, Eur J Neuroscience and Knoth et al. 2010, Plos One).
3. Lines 57-61, perhaps expand on this neurogenesis description, define integration as well
Response: Thank you for noticing the lack of detail. We have added wording to more clearly define maturation and integration. Additionally, the following three references have been added: Petreanu & Alvarez-Buylla 2002, J Neurosci; Carlen et al., 2002,Curr Biol; and Jessberger et al., 2003, Eur J Neurosci.
4. Line 63, remove “in” at end of sentence.
Response: This has been corrected.
5. Line 75, change references to “[45-48]”
Response: This has been corrected.
6. Lines 78-82, perhaps add more information about why ethologically relevant stressors are useful in the laboratory.
Response: This information has been included.
7. Lines 80-82, the sentence starting with “lastly” does not appear to be related to the first two points regarding ethologically relevant stressors.
Response: This has been corrected.
8. Lines 100 and 102, While the reference chosen, Cacioppo, et al. 2015 [57], to support the author’s assertion that social deprivation activates the HPA axis does indeed make this claim in its text, I recommend avoiding this reference as support. First of all, it is not an original research article. Also, whether or not social isolation housing causes HPA hyperactivity is a very controversial topic with many contradicting original research findings. Another review article cited later in your manuscript confirms this. From Mumtaz, et al. 2018 [72], “…effects of social isolation on HPA axis…in rats are not consistent.” The problem I have with the author’s chosen reference (i.e., [57]) is that the cited original research in Cacioppo, et al. 2015 does not actually support the statement and even contradicts what is reported in a table from that very same review article (i.e., [57]). Cacioppo, et al. 2015 write, “Studies in rats similarly suggest that chronic social isolation increases corticosterone levels when experimental animals are socially isolated from a group of same-sex rats (Djordjevic et al. 2010; Dronjak et al. 2004; Garrido et al. 2012; Zlatkovic ́ & Filipovic ́ 2012, 2013), but inconsistencies have also been observed (cf. Pournajafi-Nazarloo & Partoo 2011).” This statement is followed by a general conclusion that even appears in the abstract. The problem is that only one reference cited actually found increased corticosterone levels in isolated animals (i.e., Djordjevic, et al. 2010). They also fail to discuss how the corticosterone measurement methods differed between the studies. I recommend supporting your writing with more carefully constructed references.
Response: Thank you for pointing this out. To avoid the inclusion of a measure with such inconsistent findings, we have removed this from our paper. This has been removed both on page 4 in the first paragraph of section 2.1. as well as on page 6 in the last paragraph of section 2.1.
9. Lines 105-131, While the duration of isolation is important, especially considering that brief episodes of isolation activate the HPA axis whereas prolonged isolation housing does not necessarily, it is also important to discuss the age during which isolation occurs. Are there any differences between the impact of isolation rearing (adolescence) versus isolation housing (adulthood) on neurogenesis?
Response: This is an interesting question. Our review has exclusively focused on stressors that are experienced in adulthood and the subsequent effect of such stressors on adult neurogenesis. We have added wording in the last paragraph of the introduction (page four) to specifically highlight that the stressors discussed were presented during adulthood, rather than adolescence or even prior to adolescence. Including a discussion on the effect of stressors that were experienced during adolescence or prior to adolescence on adult neurogenesis would require a major reorganization and rewrite of the current review paper.
10. Line 131, you could add another reference here: Kozareva, et al. 2010, Hippocampus PMID: 28972669
Response: Thank you so much for suggesting this reference. However, as mentioned above the scope of our review is limited to stressors presented during adulthood.
11. Line 156, I would expect to see original research cited here.
Response: The review paper has been replaced with original research and the corresponding description has been updated.
12. Line 220, I would expect to see original research cited here instead of [42].
Response: We were not sure how to address this comment. Based on the submitted manuscript, reference # 42 refers to original research conducted by Gould et al.
42. Gould, E.; McEwen, B.S.; Tanapat, P.; Galea, L.A.; Fuchs, E. Neurogenesis in the dentate gyrus of the adult tree shrew is regulated by psychosocial stress and NMDA receptor activation. J. Neurosci. 1997, 17 (7), 2492-2498.
Please note that extensive revisions led to a change of reference numbers. The citation above is now number 51.
13. Line 239, Is “long-term psychosocial stress” the same as chronic or subchronic social defeat? These terms are confusing. With so few references it might be easier to just put the details of the social defeat methodology in the manuscript, or perhaps separately in the table that I recommend adding.
Response: Thank you for noting this. We have added/changed wording throughout the manuscript to stress the difference between subchronic (more than 1 day and up to 10 days) and chronic (more than 10 days). In addition, we hope that the table will make this distinction between subchronic and chronic clearer.
14. Line 247, I challenge the author’s no one has investigated this statement. These references investigate neurogenesis following female social defeat contrary to the author’s assertion: Mol Psychiatry. 2010 Dec;15(12):1152-63. doi: 10.1038/mp.2010.34. Epub 2010 Mar 23. Environmental enrichment requires adult neurogenesis to facilitate the recovery from psychosocial stress.
Schloesser RJ1, Lehmann M, Martinowich K, Manji HK, Herkenham M.
Horm Behav. 2015 Apr;70:30-7. doi: 10.1016/j.yhbeh.2015.01.010. Epub 2015 Feb 25.
Exposure to social defeat stress in adolescence improves the working memory and anxiety-like behavior of adult female rats with intrauterine growth restriction, independently of hippocampal neurogenesis.
Furuta M1, Ninomiya-Baba M2, Chiba S2, Funabashi T3, Akema T3, Kunugi H2.
Response: Thank you for challenging our statement, we have toned down the statement. Unfortunately, we were not able to incorporate the suggested citations for the following reasons: Schloesser et al. (2010), while mentioning females in the method section, only used males for their investigation of the impact of social defeat. Furuta et al. (2015) investigated the impact of social defeat in adolescent females rather than adult females—the focus of our review.
15. Lines 262-289, Section 3 appears to only review the effects of TMT stress on neurogenesis. This immediately following a statement that most of the research out there is about predator-related odors. Where is the review of the more common type of non-conspecific psychosocial stress?
Response: Conducting a search for predator and/or predator odor AND adult neurogenesis only yielded a very small number of relevant citations—the 5 citations mentioned in our review. We also searched the articles that had referenced these 5 citations. We did not find any additional relevant citations.
16. Line 278, which part of the previously reviewed stages of neurogenesis includes this measure of apoptotic neurons.
Response: Thank you for noticing the lack of detail. We have added wording and a corresponding reference (on page 4) to clarify this link.
17. Line 293, this is the first mention of “integration” since the introduction. Are differentiation maturation, and migration not important for “species-typical behavior”? Perhaps include an example of a species-typical behavior. Consider rewriting.
Response: Great observation. We rewrote the section.
18. Lines 295-97, consider deleting text from particulary to resolved as these words do not add anything to the paragraph and detract from the main point.
Response: Thank you for the suggestion. The section has been rewritten.
19. Lines 322-330, these lines were particularly useful. I would like to see more of this in the manuscript.
Response: Thank you for the feedback. We hope that overall polishing the manuscript and addition detail throughout our review was done in a satisfactory way.
20. Line 342, rewrite “social can”. Social what?
Response: Thank you so much for noticing that a word was missing. The missing word (interaction) has been added.
21. Line 347, non-appetitive? consider a different word such as aversive or neutral. Consider rewriting.
Response: Done
22. Line 350, agonist is a noun. In my opinion this is a poor word choice when you need an adjective to describe social defeat. Consider rewriting.
Response: Done
23. Line 354, consider rewriting, “exposed to a predator or related stimuli such as odor” to be more clear.
Response: The wording has been changed.
24. Lines 290-371, Consider writing about what is different about the conclusions drawn from research on ethologically relevant stressors and ordinary laboratory stressors regarding neurogenesis, and even other topics. For instance, social stressors usually intensify drug-seeking behavior, whereas non-social stressors often do not.
Response: Thank you for your suggestion. We have reworded the first paragraph of the discussion to follow your suggestion.